# Investigation of the Use of Non-Invasive Samples for the Molecular Detection of EHV-1 in Horses with and without Clinical Infection

**DOI:** 10.3390/pathogens11050574

**Published:** 2022-05-13

**Authors:** Danielle Price, Samantha Barnum, Jenny Mize, Nicola Pusterla

**Affiliations:** 1Steinbeck Peninsula Equine Clinics, Menlo Park Clinic, Menlo Park, CA 94028, USA; dprice@steinbeckequine.com (D.P.); jmize@steinbeckequine.com (J.M.); 2Department of Medicine and Epidemiology, School of Veterinary Medicine, University of California, Davis, CA 95616, USA; smmapes@ucdavis.edu

**Keywords:** EHV-1, outbreak, diagnostic samples, qPCR, equine

## Abstract

The purpose of this study was to explore sampling options for a reliable and logistically more feasible protocol during a large EHV-1 outbreak. Seventeen horses with clinical infection as well as nineteen healthy herdmates, all part of an EHM outbreak, were enrolled in the study. Each horse was sampled two–four times at intervals of 2–6 days during the outbreak. All samples were collected using 6′′ rayon-tipped swabs. Nasal secretions were used as the diagnostic sample of choice. Additional samples, including swabs from the muzzle/nares, swabs from the front limbs, rectal swabs, swabs of the feed bin, and swabs of the water troughs were collected as well. All swabs were tested for the presence of EHV-1 by qPCR. With the exception of two EHV-1 qPCR-positive swabs from two different horses, all remaining swabs collected from healthy herdmates tested qPCR-negative for EHV-1. For horses with clinical infection, EHV-1 was detected in 31 nasal swabs, 30 muzzle/nares swabs, 7 front limb swabs, 7 feeders, 6 water troughs and 6 rectal swabs. Not all positive muzzle/nares swabs correlated with a positive nasal swab from the same set, however, and all other positive swabs did correlate with a positive nasal swab in their respective set. The agreement between nasal swabs and muzzle/nares swabs was 74%. The sampling of non-invasive swabs from the muzzle/nares should facilitate the identification of EHV-1 shedders during an outbreak, allowing for prompt isolation and implementation of biosecurity measures.

## 1. Introduction

Equine herpesvirus-1 (EHV-1) infection occurs in horses of all ages throughout the world and typically is manifested as sporadic mild respiratory disease associated with fever, primarily in horses < 2 years of age, abortion or delivery of an infected neonatal foal, and/or neurologic disease, causing morbidity, extensive movement restrictions, and loss of life [1]. Several recent EHV-1 outbreaks resulting in equine herpesvirus-1 myeloencephalopathy (EHM) support the observation that morbidity and mortality during outbreaks are higher than in the past [2,3,4,5,6,7,8,9]. Animal health professionals involved in responding to these recent EHM outbreaks have underscored the need to improve our knowledge of this disease by the systematic collection of epidemiologic data, specifically from EHV-1 outbreaks that prominently feature EHM cases. Age, breed, sex, and vaccination status have been incriminated as possible factors associated with the development of EHM [9,10,11,12,13,14,15]. Further, long-distance transportation, corticosteroid administration, hospitalization, co-infection, and stress have been incriminated as triggering factors for the reactivation of EHV-1 [16,17,18,19].

With the increased concerns about EHV-1 neurological disease, many equine showgrounds are struggling to institute compliant protocols that reduce the risk of transmission. Updated EHV-1 vaccine protocols, increased awareness about biosecurity, daily physical monitoring, and pre-show testing are all measures that have been proposed or used during active EHM outbreaks [20]. Unfortunately, there is no contemporary information on the frequency of detection or the circulation of EHV-1 in show horses. Further, no environmental monitoring programs for EHV-1 have been validated to date. This information is essential in order to institute scientifically sound protocols with the aim of monitoring EHV-1 in a population of show horses. Therefore, the purpose of this study was to explore sampling options for a reliable and logistically more feasible protocol for a large EHV-1 outbreak.

## 2. Results

The study population was composed of 32 adult horses, 4–27 years of age (median age 12 years). There were 19 geldings and 13 mares. Various breeds were represented, including Quarter Horse (11 horses), Warmblood (10), Thoroughbred (7), and other breeds (4). Initially, 19 horses composed the healthy group, but 4 horses ended moving to the EHV-1 clinical group, as they developed clinical signs. A total of 17 horses were diagnosed with EHV-1 infection based on the presence of fever, respiratory signs, and distal limb edema (12 horses), or fever and acute onset of neurological deficits (5 horses).

A total of 94 sample sets were collected from horses with confirmed EHV-1 infection (44 sample sets) and from healthy herdmates (50 sample sets). With the exception of two EHV-1 qPCR-positive swabs from two different horses (one nasal and one muzzle/nares swab), all remaining swabs collected from healthy herdmates tested qPCR-negative for EHV-1. For horses with clinical infection, EHV-1 was detected in 31 nasal swabs, 30 muzzle/nares swabs, 7 front limb swabs, 7 feeders, 6 water troughs, and 6 rectal swabs (Table 1). Not all positive muzzle/nares swabs correlated with a positive nasal swab from the same set. All other positive swabs did correlate with a positive nasal swab in their respective set. The overall agreement between nasal swabs and the other sample types was as follows: 74% for muzzle/nares swabs, 23% for front limb swabs, 23% for feeders, 19% for water troughs, and 19% for rectal swabs. All qPCR-positive samples from healthy and sick horses were positive for both, the *gB* and the *ORF 30* (N_752_) gene. The absolute values for EHV-1 qPCR-positive nasal swabs from sick horses ranged from 2.3 to 3.0 × 10^6^ (median 1899) *gB* genes/µL of purified DNA and were not statistically different (P = 0.22) from the absolute values from muzzle/nares swabs (range from 1.5 to 4.0 × 10^5^, median 1320 *gB* genes/µL of purified DNA; Figure 1).

## 3. Discussion

Recent outbreaks of EHV-1 at national and international horse shows [4,7] have shown the complexity of managing such outbreaks and preventing the often inevitable outcome of EHM. While the collection of whole blood and respiratory secretions from horses with clinical disease for the molecular testing of EHV-1 is considered the diagnostic gold standard, and is well justified by all stakeholders, the testing of in-contact horses (exposed and non-exposed), or the repeated sample collection from infected horses, is sometimes difficult to justify, mostly because of the owners’ perception that nasal swabs are invasive and cause momentary discomfort. The purpose of this study was to explore sampling options for a reliable and logistically more feasible protocol during a large EHV-1 outbreak.

In the present study, two healthy herdmates tested qPCR-positive for EHV-1. It remains to be determined if these two horses experienced silent infection or the reactivation of latent infection with subsequent nasal shedding. Nasal shedding without clinical disease is a key feature of common respiratory viruses and is one of the greatest challenges in controlling outbreaks [20]. None of the two qPCR-positive healthy herdmates ended developing clinical disease. This was in contrast to four healthy horses with EHV-1 qPCR-negative results which ended developing EHV-1 clinical infection. Overall, 31 nasal secretions from clinically infected horses tested qPCR-positive for EHV-1 at onset of clinical disease and follow-up samples. A previous study has shown that nasal swabs were as sensitive in the qPCR detection of EHV-1 than the less well-tolerated nasopharyngeal swabs [21]. Nasal swabs are presently considered the biological sample of choice for the laboratory support of EHV-1 infection. The present study results showed that even more rostral-derived sample collection, i.e., the collection from the muzzle and nares, yielded consistent EHV-1 qPCR-positive results. There was a 74% agreement between nasal swabs and muzzle/nares swabs. Interestingly, the seven discrepant EHV-1 qPCR-positive muzzle/nares swabs were all follow-up samples collected 2–3 days after the last positive nasal swab. While amount of EHV-1 target genes between the nasal and muzzle/nares swabs were not significantly different, it appears that EHV-1 remains detectable around the muzzle and nares, even after the absence of detection within the rostral nasal passages. While this observation only applies to horses with clinical EHV-1 infection, previous studies have shown that viral loads present in nasal passages between horses with respiratory disease and subclinical shedder are often similar [22,23]. This implies that the monitoring of EHV-1 status in diseased and subclinically infected horses may be performed via both, nasal and/or muzzle/nares swabs. Additional swab types tested less frequently qPCR-positive for EHV-1 but were always associated with a positive nasal swab and/or muzzle/nares swab from the same set. Of interest was the detection of EHV-1 in the feces of 6/44 sample sets from horses with clinical infection and has to the authors’ knowledge never been demonstrated in domestic horses. The detection of herpesviruses from feces has previously been reported as a non-invasive method to document herpesvirus infection in wild equids [24]. While it may be more practical to assess a horse’s EHV-1 shedding status by sampling the environment, more work needs to be performed in order to optimize the location of the sample within the stall as well as the surface sampled. 

Study limitations related to the relatively small number of horses included in the study. Follow-up studies are needed, especially in subclinical shedders, to determine the accuracy of using less-invasive sample types to assess EHV-1 status. Further, the detection of EHV-1 by qPCR did not allow drawing any conclusions regarding the contagiousness of these horses. A recent study determined that irrespective of the environment–material evaluated (leather, polyester-cotton fabric, pinewood shavings, wheat straw, and plastic), viable virus could still be recovered at 48 h following standard inoculation with EHV-1 [25]. An additional study showed that EHV-1 remained stable and infectious in water for up to three weeks [26].

## 4. Materials and Methods

### 4.1. Study Population

The study population was composed of show and pleasure horses with confirmed clinical EHV-1 infection and healthy herdmates from a large boarding facility which experienced an EHM outbreak from 22 January to 13 March 2022. Additional EHM cases temporally and epidemiologically associated with the outbreak but from different locations admitted during the same period to the William R. Pritchard Veterinary Medical Teaching Hospital, School of Veterinary Medicine, University of California at Davis were also enrolled in the study. Clinical EHV-1 infections were defined as horses with fever, respiratory signs, and distal limb edema (EHV-1 infection), and horses with fever and acute onset of neurological signs (ataxia, weakness, proprioceptive deficits, urinary incontinence; EHM infection). All clinical EHV-1 infections were supported through EHV-1-positive qPCR testing according to previously established protocols [8,27]. Horse enrollment was voluntary and informed written consent was available for each study horse.

### 4.2. Sample Collection and Analysis

Wearing disposable gloves, the attending veterinarians collected nasal secretions from the rostral nasal passages using one 6” rayon-tipped swab (Puritan^®^ Sterile Rayon Tipped Applicators, Guilford, ME, USA). The swabs were advanced into the ventral meatus of either the right or the left nostril and allowed to soak for 10 s while gently rotating. Following collection of nasal secretions, the individual swabs were placed in 10 mL evacuated blood tubes (BD Vacutainer^®^, Franklin Lakes, NJ, USA) without any added solution. Each horse was sampled 2–4 times at intervals of 2–6 days. Additional rayon-tipped swabs were collected, including swabs from the muzzle/nares, swabs from the front limbs, rectal swabs, swabs of the feed bin and the walls adjacent to where horses were fed, and swabs of the water bucket/through rim. All samples were frozen at −80 °C for regulatory reasons and processed at the time the quarantine was lifted. 

All swabs were processed for total nucleic acid purification using an automated nucleic acid extraction system (QIAcubeHT, Germantown, MD, USA) according to the manufacturer’s recommendations. Purified nucleic acids were assayed for the presence of EHV-1 (*gB* gene assay and D/N/H_752_ allelic discrimination assays) using a combined thermocycler/fluorometer (QuantStudio 5, Applied Biosystems, Foster City, CA, USA) with the standard thermal cycling protocol: 2 min at 50 °C, 10 min at 95 °C, and 40 cycles of 15 s at 95 °C, and 60 s at 60 °C [8,21]. The PCR reactions for each assay was composed of a commercially available mastermix (Universal TaqMan Mastermix with AmpErase UNG, Applied Biosystems, Foster City, CA, USA), containing 10 mM Tris (pH 8.3), 50 mM KCl, 5 mM MgCl_2_, 300 μM each of dATP, dCTP and dGTP, 600 μM dUTP, 0.625 U of AmpliTaq Gold per reaction, 0.25 U AmpErase UNG per reaction, 400 nM of each primer and 80 nM of the respective TaqMan probe, and 1 μL of gDNA sample for a total volume of 12 μL. The qPCR-positive EHV-1 results were reported qualitatively (positive or negative) and quantitatively. Quantitative qPCR-results for EHV-1 were expressed as the number of *gB* target genes per µL of purified DNA, as previously reported [28].

### 4.3. Data Analysis

Demographic information from the EHV-1 infected horses and healthy herdmates was evaluated using descriptive analyses (mean, standard deviation, and median). Data were tested for normality using the Shapiro–Wilk test. EHV-1 viral loads for the various sample types were compared using a test of the null hypothesis (Student’s t-test or Mann–Whitney U test). All statistical analyses were performed using Stata Statistical Software (College Station, TX, USA), and statistical significance was set at *p* < 0.05. 

## 5. Conclusions

In conclusion, the study results showed that the muzzle/nares swabs showed, in comparison to the nasal swabs, good overall agreement in the detection of EHV-1 in horses with clinical disease. Non-invasive swabs from the muzzle/nares should facilitate the identification of EHV-1 shedders during an outbreak, allowing for prompt isolation and implementation of biosecurity measures, therefore reducing the overall morbidity rate and the length of the outbreak.

## Figures and Tables

**Figure 1 pathogens-11-00574-f001:**
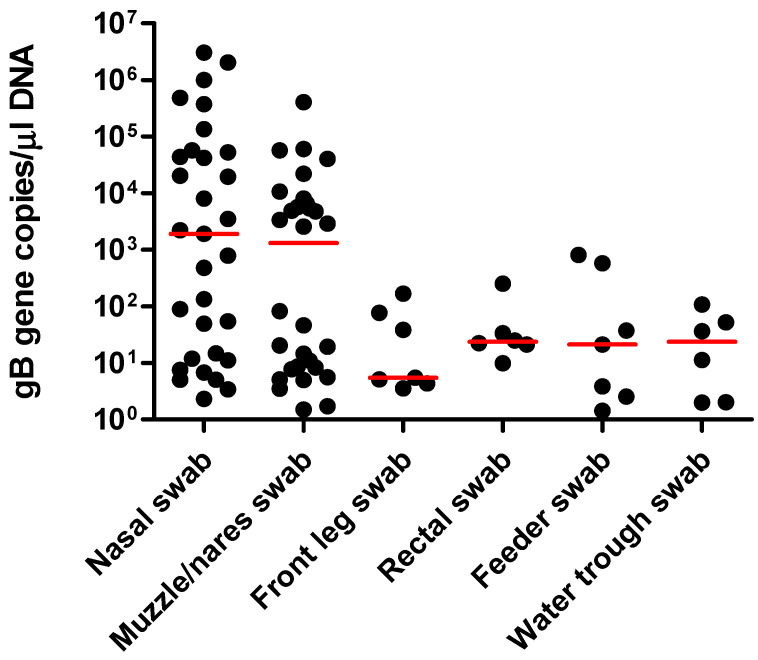
Absolute quantitation for each EHV-1 qPCR results from nasal swabs, nares/muzzle swabs, front limb swabs, rectal swabs, feeder swabs, and water trough swabs. The results are expressed as the number of target *gB* genes per µL of purified DNA. The horizontal red lines represent the median values.

**Table 1 pathogens-11-00574-t001:** qPCR results for EHV-1 in nasal swabs, muzzle/nares swabs, front limb swabs, rectal swabs, feeder swabs, water trough swabs from 32 horses with clinical EHV-1 infection, and healthy herdmates. Sample sets were collected 2–4 times at intervals of 2–6 days for each study horse. The results are reported as number of EHV-1 qPCR-positive and qPCR-negative swabs. The quantitative results (range and median in parenthesis) are expressed as number of *gB* genes per µL of purified DNA.

	Nasal	Muzzle/Nares	Front Limbs	Rectal	Feeder	Water Trough
**EHV-1 infection** (44 sample sets)	31/13	30/14	7/37	6/38	7/37	6/38
Quantiative results (*gB* genes/µL DNA)	2.3–3.0 × 10^6^ (1899)	1.5–4.0 × 10^5^ (1320)	3.5–168.9 (5.5)	9.8–250.8 (23.8)	1.4–809.3 (21.2)	1.9–107.6 (23.7)
**Healthy herdmates** (50 sample sets)	1/49	1/49	0/50	0/50	0/50	0/50
Quantitative results (*gB* genes/µL DNA)	2.3	5.3	Not applicable	Not applicable	Not applicable	Not applicable

## Data Availability

Data available on request due to privacy restrictions.

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
