# Peer review of "Investigation of the Use of Non-Invasive Samples for the Molecular Detection of EHV-1 in Horses with and without Clinical Infection"

_pathogens, 2022, doi:10.3390/pathogens11050574_

Round 1

Reviewer 1 Report

The purpose of this study was to explore sampling options for a reliable and logistically more feasible protocol during a large EHV-1 outbreak. It is important to find the least invasive tests possible for the horses (nasal swabs can be unpleasant for horses) and quick to perform for the veterinarians when more than 300 horses are gathered as during the 2021 Valence events.

However, I find the article very light for Pathogens from the point of view of methodological description but also in the description of the results.

In terms of methodology, the authors do not specify how they take their swabs (solution used, volume) before extraction. There is no description of the statistical tests performed, only the name of the software.

I disagree with the term qPCR used throughout the article. Indeed, the abbreviation qPCR describes a quantitative PCR, which requires a quantification of the viral load as described in the previous articles cited by the authors with standard curve.

In this manuscript, the authors only present Ct (Cycle threshold) values, which is not quantitative. The authors could present their data in viral loads/µl of DNA extract or viral loads/ml of recovery solution of each swab. What is the detection limit of PCR 95%?

I would like more data to better understand the results presented. For example, in table 1, I don't understand what the 2 values 31 and 13 of 31/13 correspond to, as well as the 2 values 30 and 14 of 30/14 which are not in the same direction as the following. Moreover, in the text, the authors indicate on line 64 that there are 44 "sample sets" for "confirmed EHV-1 infection" but no value of 44 in the table.

I would have liked to see the complete results of the sets in order to be able to compare the viral loads between the different biological samples for the same horse at a given time and at different times.

Author Response

The purpose of this study was to explore sampling options for a reliable and logistically more feasible protocol during a large EHV-1 outbreak. It is important to find the least invasive tests possible for the horses (nasal swabs can be unpleasant for horses) and quick to perform for the veterinarians when more than 300 horses are gathered as during the 2021 Valence events.

However, I find the article very light for Pathogens from the point of view of methodological description but also in the description of the results.

In terms of methodology, the authors do not specify how they take their swabs (solution used, volume) before extraction.

The various methodologies have been referred in the manuscript to prevent repetition from previous studies. All the swabs were collected using 6” rayon-tipped swabs, similar to diagnostic samples collected by practitioners. Each swab was placed in a red top tube and frozen immediately. Additional information pertaining to sample collection and handling of swabs was added in the manuscript.

There is no description of the statistical tests performed, only the name of the software.

The missing information has been added to the statistical methods.

I disagree with the term qPCR used throughout the article. Indeed, the abbreviation qPCR describes a quantitative PCR, which requires a quantification of the viral load as described in the previous articles cited by the authors with standard curve. In this manuscript, the authors only present Ct (Cycle threshold) values, which is not quantitative. The authors could present their data in viral loads/µl of DNA extract or viral loads/ml of recovery solution of each swab.

The data has been converted into absolute values and reported as number of target genes (gB gene) per µl of purified DNA.

I would like more data to better understand the results presented. For example, in table 1, I don't understand what the 2 values 31 and 13 of 31/13 correspond to, as well as the 2 values 30 and 14 of 30/14 which are not in the same direction as the following. Moreover, in the text, the authors indicate on line 64 that there are 44 "sample sets" for "confirmed EHV-1 infection" but no value of 44 in the table.

Table 1 has been restructured to present the data in a more concise way.

Reviewer 2 Report

This is a very interesting study about the application of the non-invasive techniques to detect EHV-1 in horses. As the authors say, this protocol could be useful and easy to perform to take decisions in a face of outbreaks.

Below I list some minor revisions that in my opinion could improve the manuscript:

  • Title: authors consider positive horses wit clinical signs as subclinical infected but they also could be inapparent carriers. Please clarify this point in the text or change the title.
  • Line 183. The authors considerer that the gold standard for detection of EHV-1 are the nasal swabs. Although have been published a good sensitivity for that type sample currently the gold standard is the nasopharyngeal swabs. Probably in a close future the good standard will be nasal or muzzle/nares swab but more studies are needed.

Author Response

This is a very interesting study about the application of the non-invasive techniques to detect EHV-1 in horses. As the authors say, this protocol could be useful and easy to perform to take decisions in a face of outbreaks.

Below I list some minor revisions that in my opinion could improve the manuscript:

  • Title: authors consider positive horses wit clinical signs as subclinical infected but they also could be inapparent carriers. Please clarify this point in the text or change the title.

The assumption form the authors was that horses with clinical signs and EHV-1 qPCR-positive test results were considered clinically infected animals. Regarding healthy horses with detectable EHV-1 qPCR as part of the outbreak could, as suggested by the reviewer, be either subclinical shedders or horses experiencing viral nasal shedding following recrudescence of a latent stage. One could argue that either way, if there is detectable viral shedding, the mechanism is via infection (endogenous or exogenous). However, to prevent any confusion, the authors have removed the word “subclinical” in the title and highlighted the two options for EHV-1 nasal shedding with absence of clinical signs in the discussion. 

  • Line 183. The authors considerer that the gold standard for detection of EHV-1 are the nasal swabs. Although have been published a good sensitivity for that type sample currently the gold standard is the nasopharyngeal swabs. Probably in a close future the good standard will be nasal or muzzle/nares swab but more studies are needed.

In the present study, secretions collected from the rostral nasal passages using rayon-tipped swabs were considered gold standard for the present study. A previous study by the authors has shown that nasal swabs collected form clinical and subclinical horses were as sensitive in the qPCR detection of EHV-1 then the less well tolerated nasopharyngeal swabs. While this may not apply to other respiratory pathogens such as equine influenza virus and Streptococcus equi ss. equi, it holds true for EHV-1. To prevent any confusion, the authors have changed “gold standard sample” to “sample of choice for the present study”.

Round 2

Reviewer 1 Report

Authors made significant effort to address my comments.

Minor correction: ligne 172, please replace "QantStudio 5" by "QuantStudio 5"